# Individual differences in trait creativity moderate the state-level mood-creativity relationship

**Mi Zhang, Fei Wang, Dan Zhang** *

Department of Psychology, Tsinghua University, Beijing, China

* dzhang@tsinghua.edu.cn

## Abstract

The relationship between mood states and state creativity has long been investigated. Exploring individual differences may provide additional important information to further our understanding of the complex mood-creativity relationship. The present study explored the state-level mood-creativity relationship from the perspective of trait creativity. We employed the experience sampling method (ESM) in a cohort of 56 college students over five consecutive days. The participants reported their state creativity on originality and usefulness dimensions at six random points between 9:00 a.m. and 11:00 p.m., along with a 10-item concurrent mood state report. Their trait creativity was measured by the Guildford Alternative Uses Test (AUT) and the Remote Associates Test (RAT). We found moderating effects of the participants' trait creativity on their state-level mood-creativity relationship. Specifically, whereas the positive correlation between positive mood state and originality of state creativity was stronger for the participants with higher AUT flexibility scores, stronger positive correlations between negative mood state and originality of state creativity were observed for individuals with higher AUT originality scores. Our findings provide evidence in support of introducing individual differences to achieve a more comprehensive understanding of the mood-creativity link. The results could be of practical value, in developing individualized mood state regulation strategies for promoting state creativity.

## Introduction

Creativity, the ability to develop novel and useful ideas, has been suggested as the key driving force behind scientific, technological, and cultural innovation [1,2]. For decades, creativity has been regarded as a relatively stable dispositional trait, and individual differences in trait creativity have been linked to other psychological traits such as personality and intelligence [3–5]. Studies have reported a multi-dimensional construct of trait creativity. One of the most well-studied dimensions is the ability of divergent thinking, which refers to the generation of multiple ideas or solutions for a single problem [6]. Another trait that recently received increasing attention is convergent thinking ability, which is associated with finding a single solution to a problem in an analytical and deductive way [7]. Although it was once seen as blocking

**Data Availability Statement:** All relevant data are within the manuscript and its Supporting Information files.

**Funding:** This work was supported by the National Natural Science Foundation of China (grant

number: 61977041 to DZ and U1736220 to DZ and FW), the National Key Research and Development Plan (grant number: 2016YFB1001200 to DZ), and the National Social Science Foundation of China (grant number: 17ZDA323 to DZ). The funders had no role in study design, data collection and analysis, decision to publish, or preparation of the manuscript.

**Competing interests:** The authors have declared that no competing interests exist.

creativity, the necessity of convergent thinking for creative production has received increasing attention. For instance, Brophy [8] found that even divergent thinkers spent most of their time doing convergent thinking during creative problem-solving tasks; the blind variation and selective retention (BVSR) theory proposed that both divergent and convergent thinking are necessary processes throughout the phases of creativity [7,9].

Creativity can also be viewed as a dynamically fluctuating state, as individuals do not always maintain their peak creative performance [10–12]. Accordingly, recent studies are starting to explore possible contextual or situational factors that influence state creativity. In particular, the relationship between mood state and state creativity has attracted the attention of many researchers [13–18]. Meta-analytical studies have shown that positive moods, such as happiness, enhance creativity, especially creative ideation, as compared to neutral or negative moods [14,15]. Several possible explanations were proposed. For example, Isen and associates proposed a theory that positive feelings on one hand facilitate extensive and diverse positive materials in memory and on the other hand influence the way in which cognitive materials are organized and related [19–21]. What's more, positive moods may increase cognitive flexibility, which in turn enhances individuals' sensitivity to novel stimuli and their divergent thinking performance [13,22–24]. Positive moods can also be a signal of a non-threatening environment and can motivate individuals to explore broadly [25–27]. Regarding negative mood states, while some studies provided evidence for inhibition of creativity by negative moods such as anger, sadness, etc. [28–30], other studies reported that negative moods promote creative performance compared to neutral moods [31–33]. Moreover, several studies have shown that negative moods exert a non-significant effect on creativity [34,35]. Recently, it has been suggested that these apparently inconsistent findings pertaining to negative mood states could be partially resolved by further differentiating the distinct functional roles of different negative moods [14,22], deconstructing the complex components of creativity [36], or decomposing creativity into different stages [18].

Exploring individual differences may provide important information to further clarify the complex mood-creativity relationship. Several recent studies are beginning to investigate these relationships and have reported promising results on the moderating roles of some trait-level factors, such as personality and emotional intelligence, on the mood-creativity link. Conner and Silvia [28] showed that people with greater amounts of the openness trait had stronger mood-creativity relationships: their self-rated everyday creativity was more likely to be affected by their daily mood states. Parke and colleagues [37] suggested that employees with higher emotion facilitation ability, which is a facet of emotional intelligence, are better at utilizing their positive moods to enhance creativity, thus strengthen the positive mood-creativity link. The reported interactions between within-participant factors (i.e., mood) and between-participant factors support the interactionist perspective that the interaction between state factors and individual differences can foster or inhibit creativity [11,38,39]. In other words, different individuals may have different yet stable mood-creativity relationships. The individual difference perspective also has practical implications: effective creativity facilitation solutions can be achieved in an individualized manner depending on each individual's trait factor scores. Nevertheless, the above-reviewed studies were limited in revealing the trait-state interplay, as possible trait-level influences were addressed only from certain specific domains.

Investigating trait creativity may provide a comprehensive and complete overview from the individual difference perspective. Trait creativity is conceptualized to describe people's creative capabilities, therefore could reflect a good combination of possible creativity-related dispositional factors. Using trait creativity measurements such as the Guildford Alternative Uses Test (AUT) [6], the Remote Associates Test (RAT) [40] and other classical tests, researchers have found significant correlations between trait creativity and personalities (especially openness

and extraversion [5]), intelligence [41], and working memory [42,43], etc. Establishing a link between the trait-level and state-level components of creativity is expected to deepen our understanding of the state-level mood-creativity relationship. Moreover, such an investigation is also of greater practical value, as the single trait creativity measurement is hypothesized to provide more effective facilitation other than any other domain-specific trait factors (e.g. openness, emotional intelligence).

The present study aimed to investigate the state-level mood-creativity relationship from an individual difference perspective using trait creativity. The experience sampling method (ESM) was employed to record state creativity and mood states in daily life situations [44–47]. Participants reported both originality (i.e., the relative rarity of creation within a given reference group) and usefulness (i.e., being comprehensible and socially meaningful) of their daily creative activities. The dissociation of originality and usefulness is expected to provide a better description of state creativity, as these two dimensions are believed to provide distinct yet complementary information about creativity [48–51]. The participants' trait creativity was measured procedure using the two classical tests of the AUT and the RAT. The AUT and RAT have different focuses, with the former focused on divergent thinking and the latter on convergent thinking [42,52,53]. As several trait factors (i.e., openness, emotion intelligence) have been reported to moderate the state-level mood-creativity relationship and trait creativity can be regarded as an integrated concept possibly covering all creativity-related trait factors, we expected to see a moderating role of trait creativity on the state-level mood-creativity relationship. Following previous studies on the trait-state interplay of creativity in relevant domains [11,28,37–39], we hypothesize to observe a stronger state-level mood-creativity relationship for individuals with higher trait creativity scores. Specifically, a more positive state-level correlation between positive mood and state creativity is expected to be found for more (trait) creative individuals. Given the controversial findings on the direction of the link between negative mood and state creativity, the moderating effect of trait creativity on this link remains to be explored.

## Materials and methods

### Participants

Fifty-six healthy Chinese undergraduates (21 females) participated in this study. The mean age of the participants was 19 years (ranging from 18 to 23). The participants were recruited on an introduction to psychology course at Tsinghua University and were given course credits for their participation. Informed consent was obtained from all individual participants included in the study. The study was conducted following the Declaration of Helsinki and its later amendments, or comparable ethical standards. Approval was given by the local Ethics Committee of Tsinghua University. To encourage compliance in administering the ESM survey, participants all had a regular daily routine and an internet-ready mobile phone. Among them, two participants failed to reach the minimum completion rate (33%) and were excluded from the analysis.

### Materials and procedures

**Trait creativity.**    A paper-and-pencil measurement composed of the Guilford AUT [6] and the RAT [40] Chinese version from Xiao, Yao, & Qiu [54] was given to each participant to assess their trait creativity before the ESM procedure.

The AUT asked examinees to list as many possible uses for common prompts as they can within 5 min. "Newspaper" and "plastic bottle" were used in the present study. Four independent coders (two for each prompt), blind to the identity of the participants, were invited to

code the answers together, with discrepancies resolved by consensus. Before scoring, two coders simplified the answers for each prompt by cutting unnecessary particles and excluding impossible uses and incomprehensible expressions. Another two coders (one for each prompt) then categorized the coded answers according to a predetermined catalog after Qun [55]. For the newspaper, the uses were categorized into graphing, stage property, encasement, weapon, filler, physical and chemical properties, cleaning, recycling, information, and others; For the plastic bottle, the uses were categorized into the container, stage property, weapon, recycling, handcraft, and others. Following Dippo's scoring procedure [56]: *AUT originality* was defined as the number of uses that occurred in less than 10% of all the answers; *AUT fluency* scores were calculated as the total number of uses in one participant's answer; *AUT flexibility* was calculated as the total number of categories. The three final component scores were obtained by calculating the sum of the z-scores for the two prompts.

The RAT asks participants to come up with a word associated with three presented words that appeared to be semantically unrelated. The Chinese version of the RAT, developed by Xiao and colleagues [54], provides standard answers for reference. The score of RAT was defined as the number of items where a participant reached a single, correct answer. In the present study, each participant had five minutes to complete the 15 items.

The participants also filled out a web-based scale to provide necessary demographic information (age, gender, etc.) and take the Raven's Advanced Progressive Matrices (RAPM) test.

**State creativity and mood states.** The experience sampling method was used to collect the participants' momentary data on state creativity and mood states. Before the study, the experimenter introduced the procedure to the participants and explained all necessary concepts, i.e., the definitions of two dimensions of creativity (originality and usefulness) and all the mood state terms. The participants installed an app called Psychorus on their mobile phones, which is a customized questionnaire platform for ESM data collection (Psychorus, HuiXin, China).

During five consecutive working days (21 of 54 participants were involved during 2017/4/10 to 2017/4/14, and the others participated during 2017/4/17 to 2017/4/21), the Psychorus app sent six questionnaire notifications per day to each participant at random time points between 9:00 a.m. and 11:00 p.m., with a minimal interval of 90 minutes. The questionnaires came with sound and vibration alerts as notifications. Upon receiving the notification, the participants were instructed to complete a questionnaire and report their mood states and state creativity related to their activities over the preceding 30 minutes (relative to the questionnaire completion start time). Participants were required to answer all questionnaires as soon as possible after each notification. In cases where participants did not immediately begin the ESM questionnaire after receiving the notification, the system would remind them every five minutes until one hour had passed, after which the push notification would disappear, and the questionnaire would close.

We designed a 12-item ESM questionnaire, with two questions for state creativity and ten for mood states. The questionnaire was kept short in order to minimize overall participant burden and improve compliance, following previous ESM studies [57,58] and especially those in the field of creativity [28,45]. The two questions for state creativity asked the participants to rate the originality and usefulness of their activities, the questions were: "During the last 30 minutes, how original/useful were your ideas or products?" Participants responded on a 100-point scale (0 = lowest, 100 = highest). The within-participant reliabilities for the everyday creativity assessments were 0.91 for originality and 0.91 for usefulness computed using the guidelines of Heck, Thomas, & Thomas [59]. The ten items for mood states were similar to the Positive and Negative Affect Schedule (PANAS) but followed a previous ESM study on affective ratings [60]. The selected ten items were relaxed, tired, happy, stressed, concentrated,

sleepy, interested, active, angry, and depressed (five positive items and five negative items). Participants reported their mood states on a 100-point scale (0 = not at all, 100 = extremely). Mood state scores from five positive items and five negative items were first averaged to achieve a combined measure of the positive affect (PA) state and the negative affect (NA) state [61,62]. The within-participant reliabilities of mood state assessment were 0.90 for PA and 0.95 for NA.

### Data analysis

Since the collected data had a nested structure, the multi-level modeling method was employed in the present study. For each state creativity (originality or usefulness), we built a two-level random model with mood states, trait creativity, and two-way interactions between them, as specified by the following equations:

$$Creativity\_s_{ij} = \beta_{0j} + \beta_{1j}PA_{ij} + \beta_{2j}NA_{ij} + \varepsilon_{ij} \tag{1}$$

$$\beta_{0j} = \gamma_{00} + \gamma_{01}AUT\_FLU_j + \gamma_{02}AUT\_FLE_j + \gamma_{03}AUT\_ORI_j + \gamma_{04}RAT_j + \mu_{0j} \tag{2}$$

$$\beta_{1j} = \gamma_{10} + \gamma_{11}AUT\_FLU_j + \gamma_{12}AUT\_FLE_j + \gamma_{13}AUT\_ORI_j + \gamma_{14}RAT_j + \mu_{1j} \tag{3}$$

$$\beta_{2j} = \gamma_{20} + \gamma_{21}AUT\_FLU_j + \gamma_{22}AUT\_FLE_j + \gamma_{23}AUT\_ORI_j + \gamma_{24}RAT_j + \mu_{2j} \tag{4}$$

*Creativity_s$_{ij}$* refers to the state creativity score (originality or usefulness) of participant *j* at the *i*-th sampling which was predicted by the simultaneous mood state at level 1. PA and NA were group-centered before entering the model. *AUT_FLU$_j$*, *AUT_FLE$_j$*, *AUT_ORI$_j$*, and *RAT$_j$* are the trait creativity scores for the three divergent thinking components (AUT fluency, AUT flexibility, and AUT originality) and convergent thinking, respectively, for participant *j*. Here $\beta_{1j}$ and $\beta_{2j}$ explicitly express the state-level mood-creativity relationship and they were dependent upon trait-level creativity as stated in (2)~(4). Significant coefficients $\gamma_{11}$ - $\gamma_{14}$ and $\gamma_{21}$ - $\gamma_{24}$ from Eqs (3) and (4) would indicate significant moderating effects of trait creativity on the relationship between mood and state creativity.

In addition, we built a similar two-level random model controlling for intelligence level (measured by RAPM; designated as *RAPM$_j$* for participant *j*):

$$Creativity\_s_{ij} = \beta_{0j} + \beta_{1j}PA_{ij} + \beta_{2j}NA_{ij} + \varepsilon_{ij} \tag{5}$$

$$\beta_{0j} = \gamma_{00} + \gamma_{01}AUT\_FLU_j + \gamma_{02}AUT\_FLE_j + \gamma_{03}AUT\_ORI_j + \gamma_{04}RAT_j + \gamma_{05}RAPM_j + \mu_{0j} \tag{6}$$

$$\beta_{1j} = \gamma_{10} + \gamma_{11}AUT\_FLU_j + \gamma_{12}AUT\_FLE_j + \gamma_{13}AUT\_ORI_j + \gamma_{14}RAT_j + \gamma_{15}RAPM_j + \mu_{1j} \tag{7}$$

$$\beta_{2j} = \gamma_{20} + \gamma_{21}AUT\_FLU_j + \gamma_{22}AUT\_FLE_j + \gamma_{23}AUT\_ORI_j + \gamma_{24}RAT_j + \gamma_{25}RAPM_j + \mu_{2j} \tag{8}$$

Note that the trait-level variables in the above models were the original trait creativity scores (and intelligence in the second model). Although significant pairwise correlations between some of these trait-level variables were observed (see Results, Table 1), the problem of multi-collinearity was not severe, with the maximal variance inflation factor (VIF) being 7.5 (less than the empirical threshold of 10).

It should also be noted that the multi-level modeling method rather than the panel analysis method with fixed effects was employed to approach the problem of cross-level interaction. The multi-level modeling method is believed to be more suitable for the consideration of the

**Table 1. Descriptive statistics and correlations among situational and individual factors.**

|  | *M* | *SD* | Range | Skew. | Kurt. | S_ori | S_use | PA | NA | RAT | AUT_FLU | AUT_FLE | AUT_ORI |
|---|---|---|---|---|---|---|---|---|---|---|---|---|---|
| S_ori | 45.62 | 23.93 | 0–100 | 0.16 | -1.01 |  | 0.29** | 0.43** | -0.12** |  |  |  |  |
| S_use | 57.17 | 23.26 | 0–100 | -0.36 | -0.97 | 0.31* |  | 0.17** | -0.12** |  |  |  |  |
| PA | 55.79 | 15.93 | 9.0–100.0 | -0.07 | -0.21 | 0.54*** | 0.34* |  | -0.57** |  |  |  |  |
| NA | 34.89 | 15.74 | 0.0–94.0 | 0.34 | -0.15 | 0.02 | -0.23 | -0.48*** |  |  |  |  |  |
| RAT | 10.33 | 2.12 | 4–14 | -0.83 | 0.77 | 0.19 | -0.01 | 0.27* | 0.00 |  |  |  |  |
| AUT_FLU | 0.36 | 1.68 | -2.42–5.14 | 0.59 | 0.32 | 0.08 | 0.09 | -0.12 | 0.11 | -0.05 |  |  |  |
| AUT_FLE | 0.24 | 1.35 | -3.50–3.10 | -0.18 | 0.28 | -0.02 | 0.07 | -0.04 | 0.20 | -0.13 | 0.45*** |  |  |
| AUT_ORI | 0.45 | 1.85 | -2.06–5.57 | 0.73 | -0.08 | -0.01 | -0.06 | -0.19 | 0.09 | -0.10 | 0.91*** | 0.29* |  |
| RAPM | 27.93 | 5.05 | 8–35 | -1.35 | 3.34 | 0.01 | -0.08 | 0.00 | -0.02 | 0.37** | 0.20 | 0.15 | 0.22 |

S_ori = state originality; S_use = state usefulness. Means (*M*), standard deviations (*SD*), and ranges of situational variables are calculated by directly aggregating all records. Correlation coefficients below the diagonal are calculated at the between-participant level, in which the state-level variables (S_ori, S_use, PA, NA) were averaged within participants and the correlated with the trait-level variables across all participants (N = 54 for the correlation analyses). Cross-sectional correlations are presented above the diagonal, in which the state-level data were pooled together across participants (N = 1317 for the correlation analyses). Pearson's correlation was used.

* $p < .05$

** $p < .01$

***$p < .001$.

research assumption of the present study. Specifically, the mood-creativity links ($\beta_{1j}$ and $\beta_{2j}$) were explicitly expressed as a random effect, emphasizing a quantitative description of these links by the participants' individual differences by their trait creativity. Such an assumption is in line with most previous studies on individual differences in the field of psychology [28,63] and allows for better generalizability of the findings beyond the sample population [64–66]. All the data analyses were performed in Mplus (version 7.4).

## Results

On average, the 54 participants completed 81.40% of all ESM questionnaires, which resulted in 1317 valid records. The valid number of records per participant ranged from 15 to 29 (the maximal number of records is 5 days × 6 per day = 30). A basic summary of the recorded data is provided in Table 1. Table 1 also shows the correlation at both between-participant (i.e. with each participant's average momentary, state-level data) and cross-sectional (i.e. with all participants' momentary, state-level data pooled together) levels. At the between-participant level, there was a negative correlation between PA and NA ($r$ = -.48, $p < .001$), and significant positive correlations between PA and the measures of state creativity (originality, $r$ = .54, $p < .001$; usefulness, $r$ = .34, $p$ = .01). There was a positive correlation between PA and RAT score ($r$ = .27, $p$ = .048) and a positive correlation between the two-state creativity ratings (originality vs. usefulness, $r$ = .31, $p$ = .02) as well. The correlations among the three components of AUT were significant, however the correlation between AUT fluency and AUT originality ($r$ = .91, $p < .001$) was much stronger than the other two (AUT flexibility and AUT originality: $r$ = .29, $p$ = .034; AUT flexibility and AUT fluency: $r$ = .45, $p < .001$). At the cross-sectional level, significant correlations were observed between the mood states (both PA and NA) and the daily state creativity (both originality and usefulness), at a similar level as the between-participant correlation results.

Table 2 shows the participant-wise correlations between trait-level and state-level creativity. The average and variance of the state-level creativity ratings (originality and usefulness) per participant were calculated and then correlated with the corresponding trait-level creativity

**Table 2. Participant-wise correlations between trait-level and state-level creativity.**

|            | AUT_FLU | AUT_FLE | AUT_ORI | RAT   |
|------------|---------|---------|---------|-------|
| M(S_ori)   | 0.08    | -0.02   | -0.01   | 0.19  |
| M(S_use)   | 0.09    | 0.07    | -0.06   | -0.01 |
| Var(S_ori) | -0.03   | 0.24    | -0.06   | -0.11 |
| Var(S_use) | -0.02   | 0.14    | -0.01   | 0.00  |

M(S_ori) is the individual mean of state creativity originality; M(S_use) is the individual mean of state creativity usefulness; Var(S_ori) is the variance of individual state creativity originality; Var(S_use) is the variance of individual state creativity usefulness. Pearson's correlation was used.

(RAT and AUT scores). Only weak and non-significant correlation coefficients were obtained, suggesting no systematic changes of one's daily state rating patterns by his/her trait-level creativity. In other words, these correlation results demonstrated sufficient independence among these variables for the subsequent multi-level analysis.

The results of multilevel modeling analyses are summarized in Table 3. Similar results were obtained with/without controlling for the Raven's Advanced Progressive Matrices (RAPM) test score (Model 1 vs. Model 2), excluding the possible confounding factor of individual intelligence difference. As Model 2 provides a more comprehensive consideration of all variables, we focus on the results from Model 2 in the following report and discussion.

**Table 3. Coefficients of the multilevel model and multilevel model with covariates.**

|                | Model 1 | | Model 2 | |
|----------------|---------|---------|---------|---------|
|                | Originality | Usefulness | Originality | Usefulness |
| B(PA)          | 0.7*** [0.62,0.79]   | 0.2** [0.08,0.33]    | 0.7*** [0.62,0.79]   | 0.2** [0.07,0.32]    |
| B(NA)          | 0.23*** [0.13,0.34]  | 0.01 [-0.1,0.12]     | 0.23*** [0.13,0.34]  | 0.01 [-0.1,0.12]     |
| B(AUT_FLU)     | 3.45* [0.58,6.32]    | 5.62* [1.95,9.29]    | 3.35* [0.69,6.01]    | 5.53* [1.86,9.19]    |
| B(AUT_FLE)     | -0.82 [-2.82,1.19]   | -0.71 [-3.16,1.75]   | -0.73 [-2.59,1.13]   | -0.63 [-3.08,1.83]   |
| B(AUT_ORI)     | -2.59 [-5.26,0.07]   | -4.84* [-8.01,-1.67] | -2.45 [-4.93,0.03]   | -4.71* [-7.87,-1.55] |
| B(RAT)         | 0.77 [-0.07,1.62]    | -0.31 [-1.49,0.88]   | 0.88 [-0.13,1.9]     | -0.2 [-1.5,1.1]      |
| B(RAPM)        |                      |                      | -0.11 [-0.82,0.6]    | -0.11 [-0.54,0.33]   |
| B(PA*AUT_FLU)  | -0.02 [-0.12,0.08]   | 0.07 [-0.12,0.25]    | -0.02 [-0.11,0.08]   | 0.06 [-0.13,0.24]    |
| B(PA*AUT_FLE)  | 0.08* [0.02,0.14]    | -0.03 [-0.13,0.07]   | 0.08* [0.02,0.14]    | -0.02 [-0.12,0.08]   |
| B(PA*AUT_ORI)  | 0.04 [-0.04,0.12]    | -0.07 [-0.22,0.08]   | 0.03 [-0.04,0.11]    | -0.05 [-0.21,0.11]   |
| B(PA*RAT)      | 0 [-0.03,0.03]       | -0.03 [-0.09,0.03]   | -0.01 [-0.04,0.03]   | -0.01 [-0.07,0.06]   |
| B(PA*RAPM)     |                      |                      | 0.01 [-0.01,0.02]    | -0.02 [-0.04,0]      |
| B(NA*AUT_FLU)  | -0.21 [-0.41,-0.02]  | -0.02 [-0.2,0.16]    | -0.21 [-0.4,-0.02]   | -0.03 [-0.22,0.17]   |
| B(NA*AUT_FLE)  | 0.05 [-0.06,0.15]    | 0.04 [-0.05,0.13]    | 0.04 [-0.06,0.15]    | 0.04 [-0.05,0.14]    |
| B(NA*AUT_ORI)  | 0.22* [0.08,0.36]    | 0.07 [-0.1,0.24]     | 0.21* [0.07,0.36]    | 0.08 [-0.1,0.26]     |
| B(NA*RAT)      | 0.07* [0.01,0.12]    | 0.02 [-0.04,0.08]    | 0.06 [0.01,0.12]     | 0.03 [-0.04,0.09]    |
| B(NA*RAPM)     |                      |                      | 0 [-0.01,0.02]       | 0 [-0.02,0.01]       |

B = unstandardized coefficients; B(PA) = $\gamma_{10}$; B(NA) = $\gamma_{20}$; B(AUT_FLU) = $\gamma_{01}$; B(AUT_FLE) = $\gamma_{02}$; B(AUT_ORI) = $\gamma_{03}$; B(RAT) = $\gamma_{04}$; B(RAPM) = $\gamma_{05}$; B(PA*AUT_FLU) = $\gamma_{11}$; B(PA*AUT_FLE) = $\gamma_{12}$; B(PA*AUT_ORI) = $\gamma_{13}$; B(PA*RAT) = $\gamma_{14}$; B(PA*RAPM) = $\gamma_{15}$; B(NA*AUT_FLU) = $\gamma_{21}$; B(NA*AUT_FLE) = $\gamma_{22}$; B(NA*AUT_ORI) = $\gamma_{23}$; B(NA*RAT) = $\gamma_{24}$; B(NA*RAPM) = $\gamma_{25}$

95% confidence intervals are shown in brackets.

* $p < .05$

** $p < .01$

*** $p < .001$.

The main effects of PA on state creativity were significant (originality: $\gamma_{10}$ = .70, $p$ < .001, 95% CI [0.62, 0.79]; usefulness: $\gamma_{10}$ = .20, $p$ = .01, 95% CI [0.07, 0.32]). The main effect of NA on state creativity was only significant for originality ($\gamma_{20}$ = .23, $p$ < .001, 95% CI [0.13, 0.34]). There were also significant influences of trait creativity on state creativity. The main effects of AUT fluency score on state creativity originality and usefulness were both significant (originality: $\gamma_{01}$ = 3.35, $p$ = .048, 95% CI [0.69, 6.01]; usefulness: $\gamma_{01}$ = .5.53, $p$ = .013, 95% CI [1.86, 9.19]). AUT originality score had a negative impact on usefulness of state creativity ($\gamma_{03}$ = -4.71, $p$ = .014, 95% CI [-7.87, -1.55]).

More importantly, significant cross-level interactions were obtained between mood states and trait creativity. The relationship between the positive mood state and the originality rating of state creativity was significantly moderated by the AUT flexibility score ($\gamma_{12}$ = .08, $p$ = .029, 95% CI [0.02, 0.14]): A stronger positive correlation was observed for individuals with higher AUT flexibility scores than for those with lower scores (Fig 1). Meanwhile, the relationship between the negative mood state and the originality rating of state creativity was significantly moderated by the AUT originality score ($\gamma_{23}$ = .21, $p$ = .015, 95% CI [0.07, 0.36]): the individuals with higher AUT originality score showed a stronger positive correlation (Fig 2). The moderating model is illustrated in Fig 3.

## Discussion

In the present study, we employed ESM to study the mood-creativity relationship in a daily life context. By calculating the correlation of self-report data on mood states and state creativity six times per day over five consecutive days, we found that individual creativity states of originality and usefulness were positively and negatively correlated with their simultaneous positive

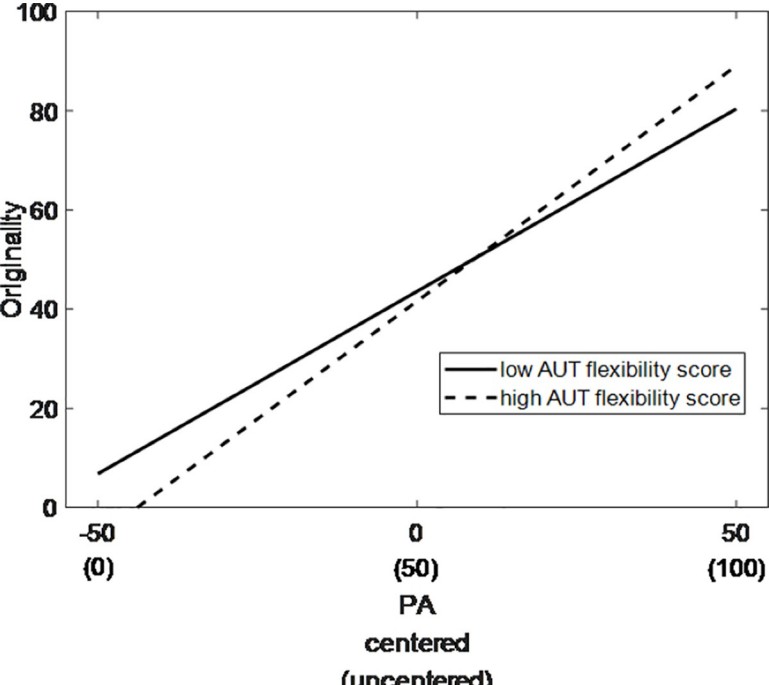

**Fig 1. Moderating effect of divergent thinking (flexibility) on state originality.** Simple slopes for the influence of positive mood state on state originality for individuals with different AUT flexibility (AUT_FLE) scores based on centered data (high = M + 1 SD, low = M– 1 SD). Units of the y-axis reflect raw scores.

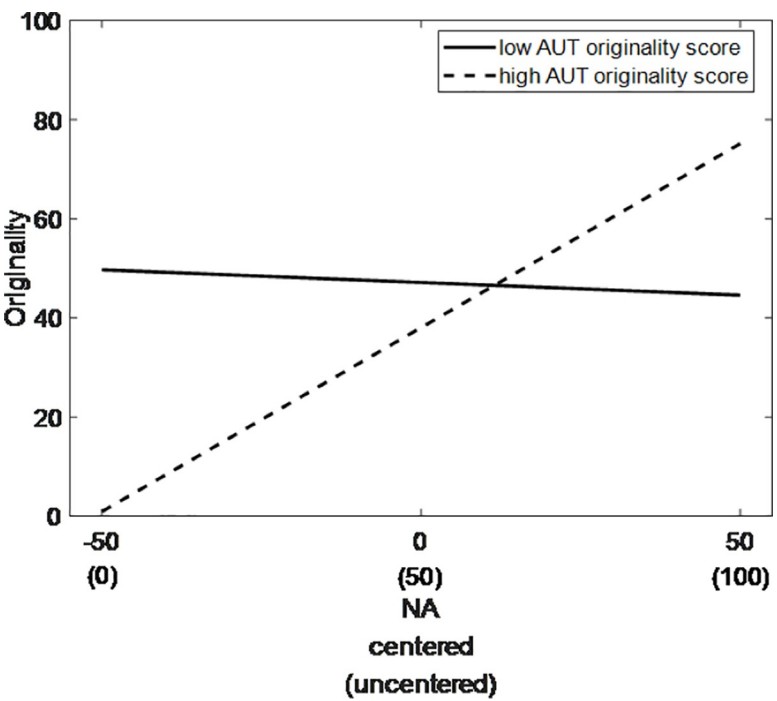

**Fig 2. Moderating effect of divergent thinking (originality) on state originality.** Simple slopes for the influence of negative mood state on state originality for individuals with different levels of AUT originality (AUT_ORI) score based on centered data (high = M + 1 SD, low = M– 1 SD). Units of the y-axis reflect raw scores.

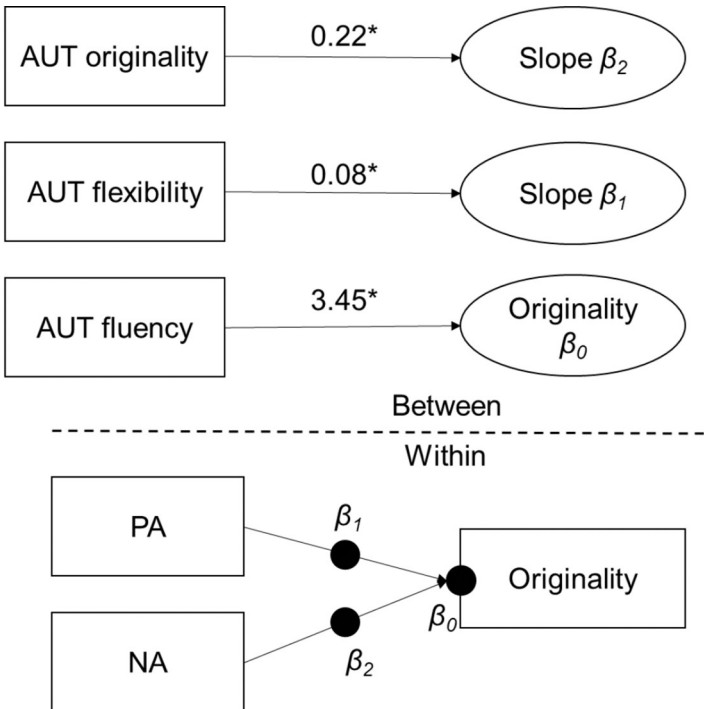

**Fig 3. Illustration of the moderating effects of trait creativity on the state-level mood-creativity correlations.** The state-level originality was significantly influenced by trait creativity of AUT fluency, the interaction between positive mood state (PA) and AUT flexibility ($\beta_1$), and the interaction between negative mood state (NA) and AUT originality ($\beta_2$). The significant interactions showed moderating effects.

and negative mood states, respectively. Importantly, we found that participants' trait creativity moderated state-level mood-creativity relationships. The positive correlation between positive mood state and originality of state creativity was stronger for the participants with higher AUT flexibility scores. More interestingly, stronger positive correlations between the negative mood state and originality of state creativity were also found for individuals with higher AUT originality scores.

By decomposing state creativity into originality and usefulness, our results contribute towards a deeper understanding of the functional role of positive mood states. While the overall positive correlation between the positive mood state and the state creativity was in good agreement with previous studies [14,15], our results showed that the positive mood states might contribute more toward state originality than toward state usefulness, as reflected by the much larger coefficient values of the positive mood states in the originality model as compared with the usefulness counterpart. The negative mood states also showed distinct contributions to the two-state creativity dimensions, with significant contributions only to state originality. As state originality is probably the most studied creative state, our results are in accordance with those that reported a promotion effect of negative moods on creative performance [31–33]. Although further studies are necessary to elucidate the underlying mechanism of these links, this observation provided support for the necessity of the decomposition of state creativity into originality and usefulness and for the first time showed the influence of mood states on the usefulness dimension of state creativity.

More importantly, we demonstrated that individual differences in trait creativity could influence these state-level mood-creativity links. For the participants with a possibly better divergent thinking capability (reflected by their AUT flexibility score), the impact of the self-referenced intensity of the positive mood states had a stronger positive influence on their state creativity of originality. While the positive link is in accordance with the mood-congruent retrieval theory and the broaden-and-build model suggesting that the positive mood state can facilitate creative processes by increasing material gathering and switching [25,67], we made further extensions on an individual difference perspective: Individuals more capable of flexibly switching between categories in divergent thinking could perform material gathering and switching more efficiently during positive mood state for enhanced creativity.

Our findings on the relationship between negative mood state and state creativity provide evidence that may help resolve the controversy from the perspective of individual differences. Whereas the originality rating of state creativity seemed to be facilitated by experiencing a relatively more negative mood state for the participants with higher divergent thinking (AUT originality) ability, weak (even negative) effects were seen for participants with lower divergent thinking ability. As a signal of a problematic situation, negative mood states are likely to indicate problems at hand, as well as the necessity of higher effort and persistence [26,68]. Under relatively high negative moods, participants with higher originality in divergent thinking abilities may benefit more from the extra effort by rejecting a familiar conventional approach and reaching for a more original idea [31–33].

The interaction effects could also be interpreted under the framework of the dual pathway to creativity model [22]. While it has been proposed that positive mood states promote creativity by facilitating cognitive flexibility and negative mood states promote creativity by increasing cognitive persistence, our results provide further evidence in support of this model for an individual difference perspective. Specifically, positive mood states could have a larger influence on individuals with higher cognitive flexibility capacities (reflected by the AUT flexibility score), leading to enhanced state creativity. Individuals with higher AUT originality scores might be associated with better cognitive organization capacities [20], therefore experiencing a larger benefit by negative mood states.

There are several limitations to the present study to be noted. First, the recruited participants were college students and they were mostly engaged in their college life on campus when their daily data were collected. To further validate the generalizability of the present findings, it would be preferred to collect data from a more diverse population, e.g. including working people in their working environment [69]. Also, here we relied on the participants' self-evaluation of their state-level mood and creativity. Given the rapid development of wearable biosensing technologies and machine learning methods [70–72], it is expected to have a momentary evaluation of one's state creativity and mood states in an objective way that could further our understanding of the mood-creativity link. Last but not least, while the mood states were categorized into positive and negative moods, it might be necessary to have a more fine-grained categorization. Specifically, it has been recently re-ignited to explore the specific impact of different kinds of fine-grained affect or mood on daily life [25,73,74].

In summary, the present study explored how individual differences in trait creativity could moderate the state-level mood-creativity relationship. The daily tracking data revealed the dependence of state-level mood-creativity correlations on an individual's trait creativity. These results not only demonstrate a complex interaction between trait creativity and the state-level mood-creativity relationship but also are of practical value in developing individualized mood state regulation strategies for promoting state creativity.

## Supporting information

**S1 File. Dataset for this study.** The datasets generated for this study.
(XLSX)

## Acknowledgments

The authors would like to thank Mr. Di Zhou, Dr. Zhen Sun, and Dr. Zhehan Jiang for statistical analysis consultancy.

## Author Contributions

**Conceptualization:** Mi Zhang, Fei Wang, Dan Zhang.

**Formal analysis:** Mi Zhang.

**Funding acquisition:** Dan Zhang.

**Project administration:** Mi Zhang.

**Writing – original draft:** Mi Zhang, Dan Zhang.

**Writing – review & editing:** Mi Zhang, Fei Wang, Dan Zhang.

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
