## [Decision Letter · Decision Letter 0]

19 Feb 2020

PONE-D-19-31431

Individual Differences in Trait Creativity Modulate the State-Level Mood-Creativity Relationship

PLOS ONE

Dear Dr. Zhang

Thank you for submitting your manuscript to PLOS ONE. After careful consideration, we feel that it has merit but does not fully meet PLOS ONE’s publication criteria as it currently stands. Therefore, we invite you to submit a revised version of the manuscript that addresses the points raised during the review process.

You will find very useful comments from the reviewers from which I want to highlight 1) the need to explain your work in constrast to the literature mentioned by revierwer 1, and 2) explain very clearly your rationale for choosing multilevel analysis instead of panel data as requested by reviewer 2. I would really apreciate an special effort to explain your findings in a much clearer way. Following the tables is very hard.

I think this paper makes a very good contribution to our understanding of creativity processes.

We would appreciate receiving your revised manuscript by may 31 . To enhance the reproducibility of your results, we recommend that if applicable you deposit your laboratory protocols in protocols.io, where a protocol can be assigned its own identifier (DOI) such that it can be cited independently in the future. For instructions see: http://journals.plos.org/plosone/s/submission-guidelines#loc-laboratory-protocols

We look forward to receiving your revised manuscript.

Kind regards,

Carlos Andres Trujillo, PhD

Academic Editor

PLOS ONE

Journal Requirements:

2. Please remove your figures from within your manuscript file, leaving only the individual TIFF/EPS image files, uploaded separately.  These will be automatically included in the reviewers’ PDF.

Reviewers' comments:

Reviewer's Responses to Questions

**Comments to the Author**

1. Is the manuscript technically sound, and do the data support the conclusions?

Reviewer #1: Yes

Reviewer #2: Partly

2. Has the statistical analysis been performed appropriately and rigorously? 

Reviewer #1: Yes

Reviewer #2: No

3. Have the authors made all data underlying the findings in their manuscript fully available?

Reviewer #1: Yes

Reviewer #2: Yes

4. Is the manuscript presented in an intelligible fashion and written in standard English?

Reviewer #1: Yes

Reviewer #2: Yes

5. Review Comments to the Author

Reviewer #1: The manuscript needs one more careful reading to address some typographical and grammatical errors. Further, it needs to give additional acknowledgement to the extensive work done by Isen and associates on the relationship between mood and creativity, much of which was subsumed in the Bass et al. meta analysis, and to comment on how the findings align with findings from DeDreu et al (2008) when it comes to how different emotional states can influence creative outputs through different channels. A more generous discussion on how findings from this work align with earlier work will affirm the current works contribution to this important research area.

Reviewer #2: In the study “individual Differences in Trait Creativity Modulate the State-Level Mood-Creativity Relationships”, the authors state that creativity, far of being a stable state, depends on contextual factors. The authors focus on mood, one factor that enhances creativity with differences according its dimensions of study (). The authors suggest that a complete understanding of this relationship should consider trait-level factors, whose relevance has been widely demonstrated in other contexts. Thus, the authors empirically explore the influence of creativity traits on the relationship state mood. The findings of the study support that creativity trait has a moderator effect differential according to the creativity state-level.

In the introduction the authors justify the importance of the study for practice. The introduction of state level creativity and mood is concrete and very well explained. Although from the introduction the importance of creativity trait is set, with the strong emphasis on the exploratory character of the study the authors miss the opportunity to make a theoretical construction. This empirical emphasis privates the authors of exploring why to understand how creativity trait from its meaning may be relevant for the academic community.

This theoretical construction is not developed in the document. The authors should note that academics are committed to the generation of knowledge, which goes beyond the focus on practical implications of trait-level factors from specific domains. (p.4).

A minor general comment looks at to standardize the used terms. For example, modulate used as synonym of moderate. If modulation is different to moderation, it is necessary to explain why. The use of modulate distracts, because moderation is the most common term in academic community. It is possible that the authors are talking about interaction, which is slightly different from moderation. However, introducing a methodological term with scarce use is confusing.

Regarding the methodology, the authors have an interesting sample of repeated measured to test the exploratory models. It is necessary to understand if the authors foresee any bias in the fact that the sample is composed of psychology students who use to be involved with the variables of the study.

Given the authors’ interest in the impact of creativity trait as moderator of state mood creativity relationship and that they have one measure of creativity trait for each of the 56 participants (repeated measures), they use multilevel analysis to test these interactions. There are some elements that deserve consideration by the authors in this regard. In multilevel methods observations are nested in contexts that influence differently their behavior. An important outcome of multilevel methods is to identify the environmental variables (2nd level) that influence individual results (1st level). In this study the observations are nested because they belong to the same individual. Although this procedure is statistically appropriate, the authors should consider explaining the bias originated in that the second-level variable (creativity trait), of course, emerges from the same individual who also reported mood and creativity-state during multiple stages of the research. In other words, there are not independence between dependent and independent variables. Furthermore, the independent variable is common for several observations.

Second, given that this study is not interested in understanding how much variance is explained at each level, how and why multilevel method provides better outcomes than a panel with fix-effects by individuals? For the previous two points I strongly recommend the authors to review any edition of Snijders and Bosker’s book.

- Snijders, T. A., & Bosker, R. J. (2011). Multilevel analysis: An introduction to basic and advanced multilevel modeling. Sage.

Regarding the results it should be noted that according to the Table 1 there are strong correlations between the variables AUT_FLU, AUT_FLE and AUT_ORI and between RAPM and RAT. However, multi-collinearity is not mention in the manuscript and the author(s) should guarantee that there are not multicollinearity effects associated with the results.

Additionally, the authors should review the notation in Table 2. There are signs in the table without interpretation in the note. Additionally, the note is confusing. It indicates Model 1 is the multilevel model without covariate, and model 2 is the model using RAMP score as covariate. However, both models include covariable and the contrast between the two models is the inclusion of RAMP as an additional covariable.

Although the graphs the authors present seem to have interesting information confirming the significant interactions of the results, the decision to focus on Model 1 (from table 2) is not explained.

The discussion section replicates the results section. Considering that the theoretical construction was not strong in the introduction, it is expected that construction to be tackled in the discussion section.

Finally, What are the conclusions of your study?

In an attempt to summarize the previous comments, I suggest to strengthen the study by developing further theoretical construction from the exploratory results and to evaluate the appropriateness of the methodological procedures, highlighting the possible weak elements and showing how the authors dealt with them.

6. PLOS authors have the option to publish the peer review history of their article (what does this mean?). If published, this will include your full peer review and any attached files.

Reviewer #1: No

Reviewer #2: No

---

## [Author Response · Author response to Decision Letter 0]

26 May 2020

The responses have been uploaded as a separate file names "Responses to Reviewers.doc". Thanks!

---

## [Decision Letter · Decision Letter 1]

10 Jul 2020

PONE-D-19-31431R1

Individual Differences in Trait Creativity Moderate the State-Level Mood-Creativity Relationship

PLOS ONE

Dear Dr. Zhang,

Thank you for submitting your manuscript to PLOS ONE. After careful consideration, we feel that it has merit but does not fully meet PLOS ONE’s publication criteria as it currently stands. Therefore, we invite you to submit a revised version of the manuscript that addresses the points raised during the review process.

Thank you for addressing the reviewers comments. Unfortunately, one of the reviewers could not revise the ne version in a timely manner, but I was able to go my self trough all the reviews and based on the reviewer 1 acceptance and my own reading I am happy to conditionally accept your manuscript based on the following minor revision:

I just need you to revise tables 1 and 2 to improve clarity. In table 1 there are 6 correlations in the upper right side that provide inconsistent information (e.g. the correlation between S_use and S_ori present two differen values). For table 2 please provide significance levels and the type of correlation calculated. Given the relatively low sample size, pearson correlation may not be appropriate, please clarify which type of correlation was used.

We look forward to receiving your revised manuscript.

Kind regards,

Carlos Andres Trujillo, PhD

Academic Editor

PLOS ONE

Reviewers' comments:

Reviewer's Responses to Questions

**Comments to the Author**

1. If the authors have adequately addressed your comments raised in a previous round of review and you feel that this manuscript is now acceptable for publication, you may indicate that here to bypass the “Comments to the Author” section, enter your conflict of interest statement in the “Confidential to Editor” section, and submit your "Accept" recommendation.

Reviewer #1: All comments have been addressed

2. Is the manuscript technically sound, and do the data support the conclusions?

Reviewer #1: Yes

3. Has the statistical analysis been performed appropriately and rigorously? 

Reviewer #1: Yes

4. Have the authors made all data underlying the findings in their manuscript fully available?

Reviewer #1: Yes

5. Is the manuscript presented in an intelligible fashion and written in standard English?

Reviewer #1: Yes

6. Review Comments to the Author

Reviewer #1: The authors were impressively diligent in addressing earlier comments and recommendations. Confirming the main effects from positive and negative affect on originality and flexibility in a single study has value.

7. PLOS authors have the option to publish the peer review history of their article (what does this mean?). If published, this will include your full peer review and any attached files.

Reviewer #1: No

---

## [Author Response · Author response to Decision Letter 1]

11 Jul 2020

See the reply file for details. Thanks!

---

## [Editor Report · Decision Letter 2]

20 Jul 2020

Individual Differences in Trait Creativity Moderate the State-Level Mood-Creativity Relationship

PONE-D-19-31431R2

Dear Dr. Zhang,

We’re pleased to inform you that your manuscript has been judged scientifically suitable for publication and will be formally accepted for publication once it meets all outstanding technical requirements.

Kind regards,

Carlos Andres Trujillo, PhD

Academic Editor

PLOS ONE
---

## [Editor Report · Acceptance letter]

22 Jul 2020

PONE-D-19-31431R2 

Individual differences in trait creativity moderate the state-level mood-creativity relationship 

Dear Dr. Zhang:

I'm pleased to inform you that your manuscript has been deemed suitable for publication in PLOS ONE. Congratulations! Your manuscript is now with our production department. 

Kind regards, 

on behalf of

Dr. Carlos Andres Trujillo 

Academic Editor

PLOS ONE